# Histopathological Verification of the Diagnostic Performance of the EU-TIRADS Classification of Thyroid Nodules—Results of a Multicenter Study Performed in a Previously Iodine-Deficient Region

**DOI:** 10.3390/jcm8111781

**Published:** 2019-10-25

**Authors:** Katarzyna Dobruch-Sobczak, Zbigniew Adamczewski, Ewelina Szczepanek-Parulska, Bartosz Migda, Kosma Woliński, Agnieszka Krauze, Piotr Prostko, Marek Ruchała, Andrzej Lewiński, Wiesław Jakubowski, Marek Dedecjus

**Affiliations:** 1Radiology Department II, Maria Sklodowska-Curie Institute- Oncology Center, 15 Wawelska St., 02-034 Warsaw, Poland; 2Department of Endocrinology and Metabolic Diseases, Medical University of Lodz, 4 Kosciuszki St., 90-419 Lodz, Poland; zbigniewadamczewski@gmail.com (Z.A.); alewin@csk.umed.lodz.pl (A.L.); 3Department of Endocrinology and Metabolic Diseases, Research Institute, Polish Mother’s Memorial Hospital – Research Institute, 281/289 Rzgowska St., 93-338 Lodz, Poland; 4Department of Endocrinology, Metabolism and Internal Medicine, Poznan University of Medical Sciences, 49 Przybyszewskiego St, 60-355 Poznan, Polandmruchala@ump.edu.pl (M.R.); 5Department of Imaging Diagnostics, Medical University of Warsaw, 61 Żwirki i Wigury St., 02-091 Warsaw, Poland; bartoszmigda@gmail.com (B.M.); a.kaczor@hotmail.com (A.K.); wsj.usg@wp.pl (W.J.); 6Statistician, 00-001 Warsaw, Poland; prostko.p@gmail.com; 7Department of Oncological Endocrinology and Nuclear Medicine, Maria Sklodowska-Curie Institute- Oncology Center, 15 Wawelska St., 02-034 Warsaw, Poland; Marek.dedecjus@gmail.com

**Keywords:** EU-TIRADS classification, thyroid nodules, histopathological verification, ultrasound

## Abstract

Background: To validate the European Thyroid Imaging and Reporting Data System EU-TIRADS classification in a multi-institutional database of thyroid nodules by analyzing the obtained scores and histopathology results. Methods: A total of 842 thyroid lesions (613 benign, 229 malignant) were identified in 428 patients (mean age 62.7 years) and scored according to EU-TIRADS, using ultrasound examination. In all tumors, histopathological verification was performed. Results: In EU-TIRADS 2 (154 nodules) all nodules were benign; in EU-TIRADS 3, only 3/93 malignancies were identified. In EU-TIRADS 4, 12/103 were malignant, and in EU-TIRADS 5 (278 benign vs. 214 malignant). The malignant nodules that would not have qualified for biopsy were: EU-TIRADS 3, 2/3 (67%) malignancies were <20 mm, in EU-TIRADS 4, 7/12 (58%) were <15 mm. In EU-TIRADS 5, 72/214 (34%) were <10 mm; in total, 81/229 (36%) malignant lesions would have been missed. The cutoff between EU-TIRADS 3/4 had sensitivity of 100%, specificity of 25.1%. Using cutoff for EU-TIRADS 5, 93.4%, 54.6%, respectively. Conclusion: The application of EU-TIRADS guidelines allowed us to achieve moderate specificity. The vast majority of malignancies in EU-TIRADS 3, 4, and 5 would not have been recommended for biopsy because having a smaller size than that proposed classification.

## 1. Introduction

The management of patients with nodular goiter and thyroid neoplasms is one of the most important problems in modern thyroidology. The appearance of a palpable nodule in the thyroid gland or focal lesions observed on images implies the need for decisions regarding further diagnostics and whether a conservative or surgical approach should be taken. The most important principle that should determine medical decision making is the safety of the patients we treat. This is reflected primarily by avoiding performing risky and aggravating medical procedures when their diagnostic or therapeutic value is less than the obtainable benefits. This means that the eligibility for diagnostic and therapeutic procedures should only apply to the patients for whom the benefits of a procedure will outweigh the risk of complications [1].

The widespread use of ultrasonography in thyroid diagnostic testing resulted in a rapid increase in the number of diagnosed asymptomatic lesions of the thyroid gland [2]. The next recommended step in diagnosing these numerous patients is fine-needle aspiration biopsy (FNAB). However, a thoughtless application of this principle may lead to a situation in which confirmatory testing of low-risk lesions with a stable ultrasound pattern is being repeated. On the other hand, in some cases, in an effort to obtain a diagnostic result, we forget about patient symptoms and signs and the ultrasound characteristics of thyroid lesions that constitute a clear indication for thyroidectomy. It should be emphasized that the performance of FNAB for all observed lesions is not prudent, although the result of cytological examination is regarded as a basis for further decision making by many clinicians [3].

A vast discrepancy exists between the number of tumors found in the thyroid and the number of tumors identified as malignant; furthermore, deaths resulting from thyroid malignancies are rare, showing the crucial role of the differentiation between benign and malignant neoplasms in the criteria for surgery [1,4,5]. Numerous studies have attempted to predict the risk of malignancy in thyroid nodules based on ultrasound patterns [6]. Frequently, these algorithms draw upon data from FNAB results or point scales that evaluate the risk of thyroid tumors being malignant [7,8,9].

Currently, in prospective studies, the most commonly modified and evaluated tool for the ultrasound-based classification of thyroid lesions is the one published by Horvath ten years ago [10]. However, for the results of this tool to be accepted as having further impact on clinical practice, they should be confirmed in trials conducted by independent research groups using their own data. Recently, the European Thyroid Association (ETA) published the novel European Thyroid Imaging and Reporting Data System (EU-TIRADS), based on the ATA (American Thyroid Association) guidelines, Korean guidelines, American Association of Clinical Endocrinologists (AACE) guidelines, and a review of recent studies [11].

The goal of our study was to validate the EU-TIRADS system in a multi-institutional database of thyroid nodules by analyzing the relationship between the obtained scores and the histopathology results. Furthermore, we aimed to evaluate the effectiveness of the EU-TIRADS scale in determining which nodules should be biopsied. To the best of our knowledge, this analysis constitutes the unique multicenter large cohort analysis of the diagnostic performance of the EU-TIRADS scale in a previously iodine-deficient region.

## 2. Materials and Methods

### 2.1. Patients

This retrospective study was approved by the institutional bioethical review board of each participating institution: the Maria Sklodowska-Curie Institute of Oncology in Warsaw (MSCI), the Medical University of Lodz (MUL), the Poznan University of Medical Sciences (PUMS) in Poland, and the Department of Imaging Diagnostics, Medical University of Warsaw (MUW) 

We reviewed the database from January 2009 to July 2018, obtaining data on 428 patients who had been admitted to our tertiary referral centers for thyroidectomy due to the following results of the Bethesda System for Reporting Thyroid Cythopathology (BSRTC): (a) suspicion of malignancy or neoplasm (BSRTC category IV–VI) or (b) nodular goiter with clinical symptoms (BSRTC category II) [12]. Patients with symptomatic purely cystic lesions were excluded from this study. The ultrasound examination of the thyroid was performed again on admission to the Department of Surgery just before thyroidectomy. A total of 842 thyroid nodules were included in this study that were diagnosed as benign or malignant on the basis of the final histopathological examination of the resected specimen.

### 2.2. Ultrasound Examination

The ultrasound examinations were conducted in the Department of Oncological Endocrinology and Nuclear Medicine of MSCI in Warsaw, the Department of Endocrinology and Metabolic Diseases of the Medical University of Lodz, the Department of Endocrinology, Metabolism, and Internal Medicine of PUMS, and the Department of Imaging Diagnostics at the Medical University of Warsaw. The neck ultrasound examinations were performed with the use of linear transducers (7–18 MHz by Aplio XG, Toshiba Medical Systems, Japan; 5–12-MHz by iU22, Philips Medical Systems, Bothell, Wash), and 5–15 MHz by the AIXPLORER system, (Supersonic Imagine, Aix-en-Provence, France).

Ultrasound (US) examinations were performed by one of five sonographers (two radiologists, three endocrinologists) with 9 to 22 years of experience in thyroid imaging. Physicians were blinded to the FNAB results and the final postsurgical verification when reassessing the US examinations and determining the EU-TIRADS score. During the US, the transverse and longitudinal planes for both the gland and the nodules were obtained while the patient was in the supine position. The anteroposterior, transverse, and longitudinal diameters of the gland and nodules were measured on frozen images during examination and were archived. The US features of the thyroid nodules were also prospectively recorded.

### 2.3. EU-TIRADS Score

Retrospectively, all of the 842 nodules were scored according to the European Thyroid Association Guidelines for Ultrasound Malignancy Risk Stratification of Thyroid Nodules in Adults (EU-TIRADS) [11]. The following are the characteristics of the different levels of classification according to this system:
EU-TIRADS 2 (benign category): purely cystic, entirely spongiform; risk of malignancy (RM): close to 0%, without a recommendation for biopsy.EU-TIRADS 3 (low-risk category): ovoid, smooth isoechoic/hyperechoic, no highly suspicious characteristics; RM: 2%–4%, recommendation for biopsy only for nodules >20 mm.EU-TIRADS 4 (intermediate-risk category): ovoid, smooth, mildly hypoechoic, no highly suspicious characteristics; RM: 6%–17%, recommendation for biopsy usually for nodules >15 mm.EU-TIRADS 5 (high-risk category): at least 1 of the following highly suspicious characteristics: irregular shape (taller-than-wide shape), irregular margins, microcalcifications, marked hypoechogenicity (and solid); RM: 26%–87%, recommendation for biopsy for nodules >10 mm.

### 2.4. Histopathological Examination

The final histopathological diagnosis was obtained after thyroidectomy for all 842 analyzed nodules. Of the 229 malignant neoplasms, papillary thyroid carcinoma (PTC) was the most common (184), while hyperplastic lesions were the most common among the benign lesions. The pathologists were blinded to the results of the ultrasound examination. Surgical specimens were immediately fixed in 10% buffered formalin. Representative sections from these specimens were processed and routinely stained with H&E (hematoxylin and eosin) for microscopic examination.

### 2.5. Statistical Analysis

The calculations were performed with Statistica 12 and in R vesion 3.6.0 (R Core Team, 2019). The presented 95% confidence intervals were obtained based on the exact test for binomial proportions, while all odds ratios (ORs) and their corresponding confidence intervals (CIs) were computed using logistic regression with a single covariate. The receiver operating characteristic (ROC) curves show the true positive rates versus the false positive rates corresponding to different size thresholds (in mm) indicating the need to perform FNAB. The area under the curve (AUC) was calculated.

### 2.6. Ethical Approval

All procedures performed were in accordance with the ethical standards of the institutional and/or national research committee and with the 1964 Helsinki declaration and its later amendments or comparable ethical standards. This article does not contain any studies with animals performed by any of the authors. 

### 2.7. Informed Consent

The informed consent was not obtained from patients to publish this report due to its retrospective character.

## 3. Results

### 3.1. Patients

A total of 842 thyroid lesions (613 benign, 229 malignant) were identified in 428 patients. Of the 229 malignant neoplasms, papillary thyroid carcinoma (PTC) was the most common (184), while hyperplastic lesions were the most common among the benign lesions. The distribution of the number of nodules per patient is presented in Table 1. Detailed histopathological results obtained in the examined group are shown in Figure 1. The cytological results were obtained for the 416 lesions. The Bethesda category distributions were: VI (95), V (89), IV (18), II (214). The mean age of the 428 patients was 62.7 years (age range 14–86 years). The mean size of the 842 thyroid nodules was 19.3 mm (standard deviation 12.8 mm, range 4–81 mm). The nodule categorization was as follows:

According to European Thyroid Imaging and Reporting Data System (EU-TIRADS): 154 nodules were EU-TIRADS category 2 (0 lesions were malignant); 93 nodules—EU-TIRADS category 3 (3/93, 3% of the lesions were malignant, the area under the curve (AUC) = 0.48); 103 nodules—EU-TIRADS category 4 (12/103, 19% of the lesions were malignant, AUC = 0.40), 492 nodules—EU-TIRADS category 5 (214/492, 43% of the lesions were malignant, AUC = 0.42). The ROC curves (Figure 2) show true positive rates versus false positive rates corresponding to different size thresholds (in mm) to perform FNAB.

### 3.2. Histopathological Examination

In Table 2, the distributions of the malignant and benign nodules in particular categories are presented. All nodules were preassigned into one of the four categories.

In the EU-TIRADS category 2, histopathological analysis revealed only benign lesions; in the EU-TIRADS category 3, only 3 malignancies were identified. In the EU-TIRADS category 5, benign tumors were also the majority (278 benign vs. 214 malignant).

Exclusion of the patients with concomitant Hashimoto’s thyroiditis or Graves’ disease, namely autoimmune thyroid disease (AITD), and patients with suspicion of primary hyperparathyroidism (PHP) on the basis of clinical and biochemical signs of hyperparathyroidism prior to surgery, slightly increased the initial cancer rate in EU-TIRADS 5 category from 43% to 46.6% (benign tumors 203, malignant 177).

### 3.3. Conventional B-Mode Ultrasound Examination and EU-TIRADS Categorization

Using the EU-TIRADS, each nodule was categorized as EU-TIRADS 2, 3, 4, or 5 on the basis of the US features described in ETA guidelines. The data in Table 3 present the statistical analysis of the B-mode parameters that had the best ability to discriminate between malignant and benign thyroid lesions and are important for EU-TIRADS evaluation.

Based on the ORs, we obtained US patterns such as irregular margins (OR = 13.82), solid or almost solid composition (OR = 9.82); hypoechoic echogenicity (OR = 5.75); taller-than-wide shape (OR = 4.86); markedly hypoechoic, microcalcifications (OR = 3.65); and macrocalcifications (OR = 1.60) as the US features most significantly associated with malignancy (Table 3). Hypoechogenicity and solid or almost solid composition were the most sensitive parameters for the detection of malignant lesions (93.9% and 92.6%, respectively) but they had low specificity (27.2% and 44%, respectively). However, the best specificity (higher than 80%) was achieved for taller-than-wide shape and irregular margins.

In Table 4, the statistical analysis of the accuracy of determining the thyroid nodule diagnosis when assigning EU-TIRADS categories is shown. When classifying nodules with EU-TIRADS ≥3 as cancers, 100% sensitivity was obtained, but the specificity was very low (25.1%). Increasing the EU-TIRADS category for the detection of malignancies resulted in a lower sensitivity but a higher specificity. If only the nodules assessed as EU-TIRADS = 5 were considered malignant, the results were as follows: sensitivity = 93.4%, specificity = 54.6%. Using the division of thyroid lesions into appropriate EU-TIRADS categories in relation to their size, we obtained the following statistical results: EU-TIRADS 3 and >20 mm in diameter, with sensitivity, specificity, and accuracy as follows (33.3%, 58.9%, 58.1%); EU-TIRADS 4 and >15 mm in diameter (41.7%, 50.5%, 49.5%, respectively); and EU-TIRADS 5 and >10 mm in diameter (66.4%, 29.5%, 45.5%, respectively).

Additionally, in Table 5, the statistical analysis according to the different cutoff values for the diameter from 5 mm to an appropriate diameter for each EU-TIRADS category determined whether a FNAB was recommended. For EU-TIRADS 3, two out of every three (67%) malignancies were smaller or equal to 20 mm, and in EU-TIRADS category 4, 7/12 (58%) were smaller than 15 mm. Additionally, in the EU-TIRADS category 5, 72/214 (34%) were smaller than 10 mm. The results of the statistical analysis indicate that if we apply a cutoff <10 mm for EU-TIRADS 3 and a cutoff >5 mm for EU-TIRADS 5, the sensitivity will increase at the expense of specificity.

## 4. Discussion

The classification of thyroid nodules should improve clinical practice and support decision making regarding biopsy and follow-up. However, the optimal classification system should reduce the number of biopsies of benign lesions, while at the same time, the percentage of malignant lesions identified should be as high as possible. Many international societies have published guidelines for diagnosing thyroid cancer that suggest the usefulness of different ultrasound classifications, i.e., in the USA, the ATA and American College of Radiology (ACR)-TIRADS classifications are commonly used. Recently, Pantano et al. compared the performances of ATA, American Association of Clinical Endocrinologists (AACE), American College of Endocrinology (ACE) and Associazione Medici Endocrinologi (AME) and ACR-TIRADS classifications for the identification of thyroid nodules with high-risk cytology and concluded that using the ATA and AACE/ACE/AME but not the ACR-TIRADS classification did not allow the accurate classification of nodules (up to 5% based on the ATA classification and up to 2.6% based on the AACE/ACE/AME system) [13]. Only the ATA classification system leaves “unclassified” a significant proportion of nodules with malignant cytology. However, the ACR-TIRADS classification system has the highest ROC-AUC for the identification of cytologically high-risk nodules.

Since the introduction of the novel EU-TIRADS classification system, a few studies have aimed to verify the diagnostic performance of this novel classification system in the clinical setting. The largest study so far validated the EU-TIRADS in a multicenter analysis performed in western Europe by Trimboli et al. [14]. It is important to mention that the patients involved in the study lived in an area encompassing Switzerland, France, and the United Kingdom, all of which are regions with an optimal iodine supply [15,16]. Similar to our analysis, the study was retrospective and included patients who qualified for thyroidectomy in four medical centers and underwent a detailed thyroid ultrasound before the surgery. Finally, the study group comprised 495 patients with 1058 nodules, of which 24.3% were malignant. The overall diagnostic accuracy for the threshold equal to four was 74.0%, with lower sensitivity and higher specificity than in the current study. Considering lesions in the EU-TIRADS category 5, the specificity was equal to 96.6%, but the sensitivity was not satisfactory (74.7%). The inclusion of nodules in EU-TIRADS category 4 increased the sensitivity (up to 93.0%); however, in fact, only 17% of lesions with a score of 4 turned out to be malignant.

A significant difference between our study and the Trimboli et al. was that in the study by Trimboli et al., nodules with a size below 5 mm was excluded while in the present study there was no dimensional selection of nodules.

To the best of our knowledge, this is unique large-cohort multicenter study conducted in a previously iodine-deficient area. Poland was regarded as iodine-deficient until obligatory salt iodination was introduced in 1997 [17]. Even in 2005 publication by the World Health Organization (WHO), Poland was still considered to be affected by mild iodine deficiency [15]. The latest information on the iodine supply based on subnational data from 2017 shows that Poland has become an area of adequate iodine intake; however, it is still recommended for pregnant and breastfeeding women to use additional iodine supplementation to achieve proper urine concentrations of iodine [18]. In such regions, the proportion of benign nodules is higher, and the risk of malignancy per detected nodule is lower than in countries characterized by sufficient iodine intake [6]. In addition, multinodular goiters and toxic nodular goiters are more frequently diagnosed [19]. The prevalence of focal thyroid lesions detected on ultrasound examinations, especially in older women, is as high as 50%–60% of the population [20].

In our previous study, we found that, similar to the features considered when determining the EU-TIRADS categories, irregular margins, solid or almost solid composition, hypoechogenicity or marked hypoechogenicity, taller-than-wide shape, microcalcifications, and macrocalcifications independently predicted the risk of malignancy. In addition, ill-defined margins were the single feature with the highest OR (10.77). When combining these features with sonoelastography (Asteria 3, 4 scores), the risk of malignancy increased even further. We found that sonoelastography increased the predicted risk of malignancy in nodules in TIRADS category 4; the Asteria category 4 as a solitary feature in a solid tumor could result in its categorization in TIRADS category 4, but the coexistence of high-risk features allows it to be upgraded to TIRADS category 5 [21]. In another previous analysis including 393 thyroid lesions aiming to stratify the usefulness of particular features on conventional ultrasonography and sonoelastography, hypoechogenicity and irregular margins were those with the highest predictive value (OR 10.9 and 7.5, respectively) [6], while in a meta-analysis of prospective studies, the “taller than wide” appearance had the highest OR among the features on conventional ultrasonography [22]. However, the best predictive factor regarding the risk of malignancy was the elasticity of the nodule [6]. Thus, the lack of involvement of elasticity in the EU-TIRADS score might be considered one of the limitations of the classification. Cantisani V et al. demonstrated that assessing the SR (strain ratio) with a cutoff of 2.09, increased sensitivity and specificity more than 90% in Bethesda III nodules and could be integrated in an US preoperative assessment to reduce unnecessary thyroidectomy [23].

In this multicenter study evaluating 842 thyroid lesions, we validated the EU-TIRADS ultrasound features concordant with the ETA guidelines and assessed the diagnostic performance of the proposed EU-TIRADS classification system [11]. In our study, we assigned the nodules based on ultrasound features into the EU-TIRADS categories according to the ETA guidelines [11]. We adopted three potential cut-off values for the EU-TIRADS categories and observed the elevation of the malignancy risk of the thyroid nodules, with very high sensitivity (93.4%), but only moderately satisfactory (54.6%) specificity for EU-TIRADS category 5. Using a cutoff greater than or equal to 3 on the EU-TIRADS scale, all malignancies were detected, but the highest percentage of benign lesions would have qualified for pathological verification (75%).

We also found that, if we analyzed specific subgroups of the lesions in EU-TIRADS categories 3, 4, and 5 and analyzed the indication for biopsy depending on their size, 81/229 (36%) malignant lesions would not have qualified for biopsy. Middleton et al. compared the ACR-TIRADS, ATA, and Korean Society of Thyroid Radiology (KSThR) guidelines with regard to the indication for FNAB or the follow-up of nodules with diameters larger than 1 cm [24]. The ACR performs well when compared with other guidelines; 13.9% of nodules could not be categorized using the ATA guidelines, and 9.4% of these uncategorized nodules were malignant. With the use of the ACR TIRADS, biopsy would not have been recommended for 31.8% of malignant tumors, which is similar to our results. The goal of the classification systems is not to omit clinically significant cancers. In our dataset, the vast majority of the malignancies in EU-TIRADS categories 3 and 4 would not have qualified for biopsy because they had smaller sizes than that proposed in the classification system. The following types of cancers were missed according to the histopathological results: papillary, follicular, and medullary carcinomas. The highest number of false negative results was in EU-TIRADS category 5 (72/214 malignancies). In our study, especially in EU-TIRADS categories 3 and 4, we observed an inverse dependence between the size of the lesions and the risk of malignancy. In our country, the local guidelines recommend performing biopsy of lesions that are 5–10 mm in diameter if suspicious features are observed on ultrasound [12]. If we applied this criterion in our cohort, we would have missed only 7/214 malignancies.

Ha EJ at al. recently compared the diagnostic performance of the 2015 ATA guidelines with those of the 2016 Korean Thyroid Association (KTA)/KSThR and 2017 ACR guidelines and found that when using the ATA guidelines, the unnecessary FNAB rate was high (51.2%) compared to the other guidelines [25]. Additionally, of all nodules, 7.6% did not meet the criteria proposed by the ATA guidelines. Another study published by Phuttharak et al. focused mainly on the analysis of interobserver agreement for particular TIRADS variants and was performed on a group of 108 lesions in 95 patients [26]. According to their results, the highest agreement was achieved for ACR-TIRADS, followed by the EU-TIRADS, and finally by the Siriaj-TIRADS. In contrast, the highest diagnostic accuracy was achieved using the Siraj-TIRADS, followed by the EU-TIRADS, and finally by the ACR-TIRADS. For the EU-TIRADS, a diagnostic accuracy of 72.2 or 71.3% (depending on the sonographers; for the threshold equal to 5) was achieved with substantial interobserver differences concerning sensitivity (45.8% vs. 66.7%, respectively) and lower variability in the case of specificity (79.8% vs. 72.6%, respectively). This demonstrates significant differences between different guidelines, leading to various degrees of sensitivity and specificity in the detection of thyroid cancer. The best approach is still unclear.

The only data from the Polish population can be drawn from the first single-center study assessing the risk of malignancy with histopathology as a reference diagnosis. The study included 52 patients with 140 nodules [27]. The authors yielded the following values for sensitivity, specificity, positive predictive value (PPV), and negative predictive value (NPV): 75%, 94.1%, 75%, and 94.1%, respectively. These results were obtained under the circumstance that EU-TIRADS category 4 or more was regarded as a criterion for the suspicion of cancer. The diagnostic performance obtained in previous studies aiming to evaluate the diagnostic utility of the EU-TIRADS scale differs quite significantly due to the different populations analyzed, with different percentages of malignant nodules and different iodine statuses of the population. Some of the studies analyzed selected groups of patients, i.e., only those demonstrating increased radiotracer uptake on a positron emission tomography(PET) scan. In our cohort of patients, if compared to the only previous Polish cohort described, the sensitivity obtained was much higher (98.7%), with much lower specificity (39.8%), unacceptably low PPV (38%), and the highest negative predictive value (98.8%). Thus, if a lesion is categorized as EU-TIRADS category 3 or less, the likelihood of it being malignant is very low. This might be explained by the different group sizes and the threshold size used to determine whether lesions qualify for biopsy. Selection bias might not be the cause, as in both studies, the examined patients qualified for thyroidectomy due to medical reasons and were admitted to the surgical department. Similarly, in both studies, the selection bias (surgical patients, not the general population) and the method of final verification of the character of the lesions (histopathological examination) were similar. In our research, the threshold size indicating that the lesion should be biopsied was 5 mm; however, that was not described in the paper by Skowronska et al. [27].

The benefit of the EU-TIRADS scale is its simplicity, and it was designed to make it simple to assess the risk of malignancy in clinical practice and to reduce the number of FNAB procedures in subjects presenting with a low risk of malignancy. Another problematic issue concerning the use of the EU-TIRADS to determine whether a lesion should be biopsied is the size threshold of the thyroid lesion. According to the novel ETA guidelines, lesions smaller than 10 mm in diameter should only be observed [11]. According to our data, refraining from further diagnostics for lesions smaller than 1 cm would have resulted in overlooking 72 (15%) thyroid cancers in EU-TIRADS category 5. A very good prognosis for thyroid cancer, especially papillary cancer, may discourage overtreatment. However, there is some data suggesting that even small thyroid cancer lesions might have unfavorable prognoses [28], which might constitute an argument for providing active diagnostics in cases of suspicious lesions that are smaller than 10 mm. Strictly following the guidelines, no thyroid cancer would be detected in the pT1a stage. This seems less than credible, as there is growing evidence regarding the possibility of aggressive behavior in lesions that are 5 mm in diameter or more [29]. If we include in the indication for FNAB the lesions >5 mm, it would lead to an increase in sensitivity from 66.4% to 97.2%. However, a significant decrease in specificity would occur. We must bear in mind that the EU-TIRADS classification system was designed to qualify patients for further diagnostic procedures, not for surgery. Thus, taking into account our results, we would advise the individualization of the decision to perform the FNAB procedure and consider performing FNAB in patients with lesions between 5–10 mm in diameter that present highly suspicious features on thyroid ultrasound. An additional tool that may help in the decision in these cases might be sonoelastography [21]. The decision might also be supported by additional clinical factors affecting worse prognosis, such as sex, age, and family history, when making a decision regarding patients with suspected lesions 5–10 mm in size. Our study based on Polish patients demonstrated that 36% of malignant lesions would not have qualified for biopsy using the EU-TIRADS, which is not an acceptable number of patients. Ultimately, the prognosis of thyroid cancer depends not on the size of the lesion but on the genetically determined malignant potential [30]. In the future, molecular markers might be established that would help identify the nodules that should be observed and those that should be more aggressively treated.

In a recent paper by Trimboli et al., the authors evaluated the potential role of 18F-Fluorodeoxyglucose(FDG)PET/CT in the risk stratification of thyroid nodules suspected of malignancy [31]. A PET/CT scan might increase the negative predictive value of the EU-TIRADS scale and allow a 41% reduction in the number of biopsies, while only one case of thyroid cancer would be missed. In another paper, Trimboli et al. analyzed the risk of malignancy in incidentally detected lesions, demonstrating increased focal uptake on PET/CT scans and applying the EU-TIRADS scale for risk stratification [32]. The authors conclude that nodules with increased tracer uptake on FDG PET/CT and a high EU-TIRADS category are much more likely to be malignant than lesions with only increased tracer uptake on FDG PET/CT.

In cases of indeterminate ultrasound and/or cytological results, very often patients are referred to diagnostic surgery. Supportive methods are necessary to avoid the operation and its potential complications. The first gene expression classifier was introduced to a preoperative diagnostic of thyroid nodules in the USA. Currently, the use of molecular tests is a routine clinical practice in the USA and recommended by American scientific societies. A recent study provides several genetic mutations considered markers of aggressive tumor behavior, including mutations in RAS, PIK3CA, PTEN, P53, ALK, and BRAF genes. However, FNAB is to date a reference standard in directing patients for operation and surveillance [33,34].

Finally, the limitations and strengths of the present study should be discussed. There are several limitations of the study, including its retrospective nature. The multicenter character of the research allowed for inclusion of one of the largest cohort of patients to date. Also, study population differs a lot from an unselected group of patients. However, five different specialists assessed the lesions and classified the nodules in particular EU-TIRADS categories.

Another limitation of our study is the lack of cytological verification of all nodules before surgery. This is the specific situation in patients from previously iodine deficient area that incidence of multinodular goitre is high and in patients i.e., with concomitant 7 or 8 nodules, all were assessed by ultrasound however, only 2–3 dominant (most suspected nodules) were verified cytologically before surgery. We have not included in the study patients with diffuse thyroid pathology with no concomitant thyroid nodules (Hashimoto’s thyroiditis, Graves’ disease, subacute thyroiditis). The only situation where we included such patients was in the presence of concomitant nodules (nodular variant of Hashimoto’s thyroiditis or Graves’ disease). Similarly, patients with parathyroid adenomas imitating thyroid nodules and accompanied by multinodular goiter were included. The initial cancer rate in our group of patients with EU-TIRADS 5 category was 43.5%, but after exclusion of the nodules with concomitant autoimmune thyroid disease AITD and patients with suspicion of primary hyperparathyroidism (PHP), the cancer rate was equal to 46,6% (estimated cancer rate in EU-TIRADS category 5 should be between 26%–87%).

## 5. Conclusions

In conclusion, EU-TIRADS is an easy to apply novel classification system for thyroid lesions, which might be helpful in clinical practice for the initial assessment of the thyroid lesion risk of malignancy. Our study demonstrated that the largest benefit of the scale lies in its negative predictive value and ability to identify nodules with very low risk of malignancy that could be successfully followed-up without any invasive procedures.

## Figures and Tables

**Figure 1 jcm-08-01781-f001:**
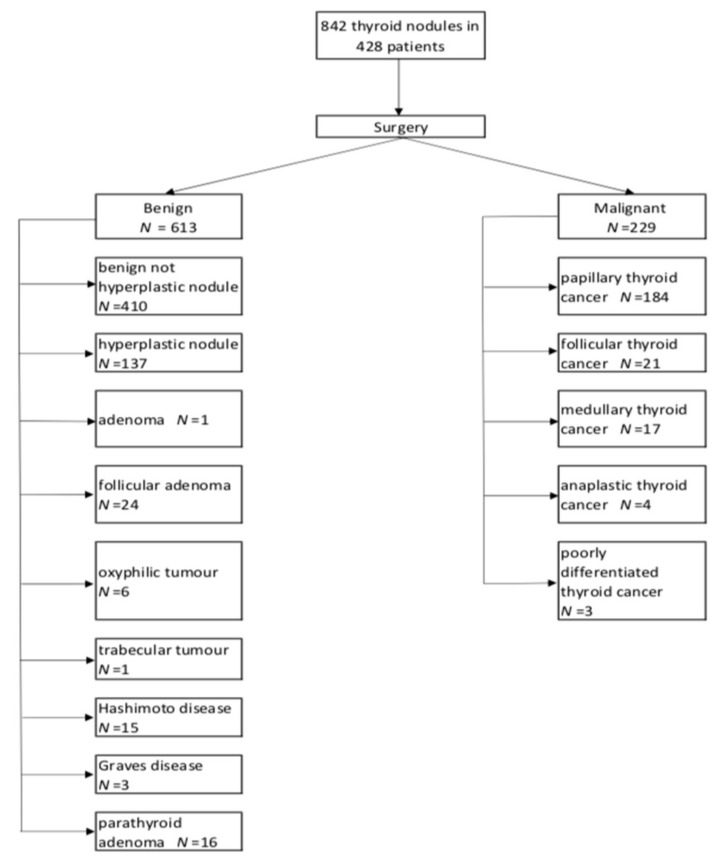
Results of histopathological examination of the thyroid nodules.

**Figure 2 jcm-08-01781-f002:**
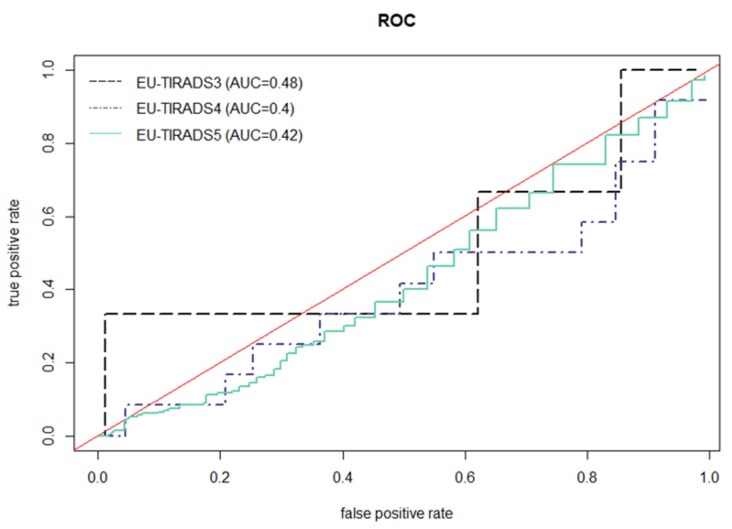
The receiver operating characteristic (ROC) curves show true positive rates versus false positive rates corresponding to different size thresholds (in mm) to perform fine needle aspiration biopsy (FNAB).

**Table 1 jcm-08-01781-t001:** Distribution of the number of nodules per patient, reflecting the degree of multinodularity.

	Nodules	Patients
Number of nodules found per patient	1	231
	2	112
	3	24
	4	26
	5	14
	6	9
	7	9
	8	3
In total	842	428

**Table 2 jcm-08-01781-t002:** Distribution of the thyroid lesions in selected EU-TIRADS categories according to histopathological results.

EU-TIRADS Category	Benign Lesions (%)	Malignant Lesions (%)
2	154 (100)	-
3	90 (97)	3 (3)
4	91 (81)	12 (19)
5	278 (57)	214 (43)

**Table 3 jcm-08-01781-t003:** Statistical analysis of B-mode parameters discriminating malignant from benign thyroid lesions (PPV—positive predictive value, NPV—negative predictive value, OR—odds ratio).

Ultrasound Feature	Characteristics% (95% CI)	Sensitivity (%)	Specificity (%)	PPV (%)	NPV (%)	OR
Echogenicity	Markedly hypoechoic	59.0 (52.3–65.4)	71.8 (68.0–75.3)	43.8 (38.2–49.6)	82.4 (78.9–85.5)	3.65 (2.66–5.01)
	Hypoechoic	93.9 (90.0–96.6)	27.2 (23.8–31.0)	32.5 (29.0–36.2)	92.3 (87.4–95.7)	5.75 (3.25–10.16)
	Isoechoic	5.2 (2.7–9.0)	76.2 (72.6–79.5)	7.6 (4.0–12.9)	68.3 (64.6–71.8)	0.18 (0.10–0.33)
	Hyperechoic	0.9 (0.1–3.1)	96.9 (95.2–98.1)	9.5 (1.2–30.4)	72.4 (69.2–75.4)	0.28 (0.06–1.19)
Margins	Irregular	75.5 (69.5–81.0)	81.7 (78.4–84.7)	60.7 (54.8–66.4)	89.9 (87.1–92.3)	13.82 (9.6–19.9)
Microcalcifications	Yes	53.7 (47.0–60.3)	75.9 (72.3–79.2)	45.4 (39.4–51.5)	81.4 (78.0–84.5)	3.65 (2.65–5.02)
Macrocalcifications	Yes	22.3 (17.1–28.2)	84.8 (81.7–87.6)	35.4 (27.6–43.8)	74.5 (71.1–77.7)	1.60 (1.09–2.35)
Composition	Solid/Almost solid	92.6 (88.4–95.6)	44.0 (40.1–48.1)	38.2 (34.1–42.4)	94.1 (90.7–96.5)	9.82 (5.84–16.50)
	Solid-cystic	7.4 (4.4–11.6)	59.7 (55.7–63.6)	6.4 (3.8–10.1)	63.3 (59.2–67.3)	0.12 (0.07–0.20)
	Cystic	0.0 (0.0–1.6)	99.7 (98.8–100.0)	0.0 (0.0–84.2)	72.7 (69.6–75.7)	NA
	Spongiform	0.0 (0.0–1.6)	96.6 (94.8–97.9)	0.0 (0.0–16.1)	72.1 (68.9–75.2)	NA
Shape	Taller-than-wide shape	45.9 (39.3–52.5)	85.2 (82.1–87.9)	53.6 (46.3–60.7)	80.8 (77.6–83.8)	4.86 (3.45–6.84)

**Table 4 jcm-08-01781-t004:** Statistical analysis discriminating accuracy of malignant and benign thyroid nodule diagnosis when assigning the EU-TIRADS category cut-off (PPV—positive predictive value, NPV—negative predictive value, OR—odds ratio). Not applicable (NA) is presented to avoid dividing by zero.

EU-TIRADS Score		Positive	Negative		Value	CI (95%)
≥3	Malignant	229	0	Sensitivity (%)	100.0	98.4–100.0
	Benign	459	154	Specificity (%)	25.1	21.7–28.8
				Accuracy (%)	45.5	42.1–48.9
				PPV (%)	33.3	29.8–36.9
				NPV (%)	100	97.6–100.0
				OR	NA	NA
≥4	Malignant	226	3	Sensitivity (%)	98.7	96.2–99.7
	Benign	369	244	Specificity (%)	39.8	35.9–43.8
				Accuracy (%)	55.8	52.4–59.2
				PPV (%)	38	34.1–42.0
				NPV (%)	98.8	96.5–99.7
				OR	49.8	15.8–157.4
5	Malignant	214	15	Sensitivity (%)	93.4	89.4–96.3
	Benign	278	335	Specificity (%)	54.6	50.6–58.6
				Accuracy (%)	65.2	61.9–68.4
				PPV (%)	43.5	39.1–48.0
				NPV (%)	95.7	93–97.6
				OR	17.2	9.95–29.71

**Table 5 jcm-08-01781-t005:** Statistical analysis with the classification of thyroid lesions into appropriate EU-TIRADS categories and in relation to their size and malignancy risk. Not applicable (NA) is presented to avoid dividing by zero.

EU- TIRADS Score	Diameter (mm)	Benign	Malignant	Accuracy (%) (95% CI)	Sensitivity (%) (95% CI)	Specificity (%) (95% CI)	PPV (%) (95% CI)	NPV (%) (95% CI)	OR (95% CI)
**3**	≤20	53	2	58.1 (47.4–68.2)	33.3 (0.8–90.6)	58.9 (48–69.2)	2.6 (0.1–13.8)	96.4 (87.5–99.6)	0.72 (0.06–8.19)
	>20	37	1						
	≤15	38	2	41.9 (31.8–52.6)	33.3 (0.8–90.6)	42.2 (31.9–53.1)	1.9 (0–10.1)	95.0 (83.1–99.4)	0.37 (0.03–4.18)
	>15	52	1						
	≤10	17	1	20.4 (12.8–30.1)	66.7 (9.4–99.2)	18.9 (11.4–28.5)	2.7 (0.3–9.3)	94.4 (72.7–99.9)	0.47 (0.04–5.44)
	>10	73	2						
	≤5	0	0	3.2 (0.67–9.14)	100 (29.24–100.0)	0 (0.0–4.02)	3.2 (0.67–9.14)	NA	NA
	>5	90	3						
**4**	≤15	46	7	49.5 (39.5–59.5)	41.7 (15.2–72.3)	50.5 (39.9–61.2)	10.0 (3.3–21.8)	86.8 (74.7–94.5)	0.73 (0.22–2.47)
	>15	45	5						
	≤10	19	5	25.2 (17.2–34.8)	58.3 (27.7–84.8)	20.9 (13.1–30.7)	8.9 (3.6–17.4)	79.2 (57.8–92.9)	0.37 (0.11–1.29)
	>10	72	7						
	≤5	0	1	10.7 (5.5–18.3)	91.7 (61.5–99.8)	0.0 (0.0–4.0)	10.8 (5.5–18.5)	0.0 (0.0–97.5)	NA
	>5	91	11						
**5**	≤10	82	72	45.5 (41.1–50.0)	66.4 (59.6–72.7)	29.5 (24.2–35.2)	42.0 (36.7–47.5)	53.2 (45–61.3)	0.83 (0.56–1.21)
	>10	196	142						
	≤5	8	6	43.9 (39.5–48.4)	97.2 (94.0–99.0)	2.9 (1.3–5.6)	43.5 (39.0–48.1)	57.1 (28.9–82.3)	1.03 (0.35–3.01)
	>5	270	208						

PPV—positive predictive value, NPV—negative predictive value, OR—odds ratio.

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
