# Peer review of "Histopathological Verification of the Diagnostic Performance of the EU-TIRADS Classification of Thyroid Nodules—Results of a Multicenter Study Performed in a Previously Iodine-Deficient Region"

_jcm, 2019, doi:10.3390/jcm8111781_

Round 1
Reviewer 1 Report
I read with interest this study due to the relevance of the topic. Really, other studies with quite similar results have been reported on this topic. However, the present study offers the possibility of evaluating EU-TIRADS perfomance in a specific region with previous iodine deficiency.
The main criticism is that the rate of malignancy recorded in class II, III, and IV is perfectly in line with that reported by previous studies (Trimboli 2018, Shen 2019), while the cancer rate of high-risk class (EU-TIRADS 5) is too lower with respect the other studies (43% vs. 87 and 81%, respectively). This is probably due to a selection bias. In fact, I can notice in Figure 1 that there were many nodules with benign histology in which the diagnosis is unclear. In example, what do the authors mean with "benign non hyperplastic nodule (n=410")? This subgroup could include a large number of lesions with unclear US presentation and probably misleading features (marked hypoechogenicity, irregular margins). Including these lesions could lead to a large number of false-positive US class (ie, EU-TIRADS 5). Also, this subgroup could include very small nodules which one observer might classify as EU-TIRADS 5 due to the difficult US examination. I recommend to make an effort to reduce as much as possible the false positive of EU-TIRADS 5. A minimum size of lesions as inclusion criterion can help in this context. In addition, I suggest to exclude completely those patients undergone thyroidectomy for thyroiditis, Graves' or parathyroid adenoma (these are probably other false positive US).
A careful reading is needed. I have noticed several typos.
Author Response
Thank you for your comments about the low percentage of malignant lesions in EU-TIRADS 5 category.
We believe that the relatively low percentage of malignant lesions reflects the relatively low incidence of malignant thyroid nodules in Polish population and the specificity of the regions with high incidence of non-malignant thyroid lesions and iodine-deficiency. We clarified this in the discussion section (line 263-266)
Thank you for your comments about “benign non hyperplastic nodule (n=410)”.
We analyzed this group of false positive nodules and concluded they were benign colloid nodules. About 30% were in the EU-TIRADS 5 category, and 41% were between 5-10 mm. We very carefully reassessed this group of nodules, using archived avi films and B-mode frozen images.
We have not included in the study those patients with diffuse thyroid pathology with no concomitant thyroid nodules (Hashimoto’s thyroiditis, Graves’ disease, subacute thyroiditis). The only situation where we included such patients was in the presence of concomitant nodules (nodular variant of Hashimoto’s thyroiditis or Graves’ disease).
Similarly, patients with parathyroid adenomas imitating thyroid nodules and accompanied by multinodular goiter were included.
We added the relevant text to the study limitations section (line 394-397).

Reviewer 2 Report
The paper titled “Histopathological verification of the diagnostic performance of the Eu-Tirads classification of thyroid nodules - results of a multicenter study performed in a previously iodine-deficient region” from Dobruch-Sobczak K. and coworkers is well written and evaluates a largely discussed topic in thyroid literature. However, the following comments should be addressed:
Major comments
In “Materials and Methods” section, the authors assert that all nodules were retrospectively scored according to the EU-TIRADS; this represents the major limitation of the study, since it is not possible to be sure that thyroid US had been correctly performed in line with EU-TIRADS recommendations, for example in terms of nodule’s features to be reported. The author should deeply discuss this aspect; 
 The study population includes patients admitted for thyroidectomy due to belonging to catergory IV-VI or II according to the Bethesda System for Reporting Thyroid Cythopathology; the paper lacks a table summurizing the results of FNA; furthermore, details on FNA should be added in both “Results” and “Discussion”; 
Minor comments
In the “Discussion” the author describes the role of sonoelastography in predicting the risk of malignancy in nodules in TIRADS category 4 and the importance of involving elasticity in the EU-TIRADS score. Recently, strain ratio measurements have been demonstrated to improve differentiation of thyroid nodules with indeterminate cytology; thus, strain ratio may help to better select patients with Thy3 nodules candidate for surgery (Cantisani et al. European Radiology 2016); please, add to the discussion; Recently, many efforts have been made to improve the diagnostic work-up of thyroid nodules; authors should also mention the role of citology and molecular markers as well as the role of elastography.
Author Response
Thank you for both remarks about the limitations of our study.
We retrospectively scored the nodules according to the EU-TIRADS; but very carefully reassessed this group of nodules using archived avi films and B-mode frozen images.
An experienced specialist in radiology and endocrinology with more than 7 years of experience in US examination collected the data and analyzed the films.
Another limitation of our study is lack of cytological verification of all nodules before surgery. The cytological results were obtained from 416 lesions. The Bethesda category distributions were: VI (95), V (89), IV (18), II (214). This is the specific situation in patients from previously iodine deficient area that incidence of multinodular goitre is high and in patients i.e. with concomitant 7 or 8 nodules, all were assessed by ultrasound however, only 2-3 dominant (most suspected nodules) were verified cytologically before surgery. The aim of this paper was to evaluate the lesions and compare it with histopathological examination as a gold standard for diagnosis. We did not want to make our results less clear by inclusion the results of cytological examinations.
Data about the FNAB were added into results and discussion (line 161-162, 389-393).
Thank you for your minor comments.
We read, with interest, the paper about SR in patient with Thy3 nodules candidate for surgery and added the following summary of the Cantisani study to the discussion:
Cantisani V et al demonstrated that assessing the SR (strain ratio) with a cut off of 2.09 increased sensitivity and specificity more than 90% in Bethesda III nodules and could be integrated in US preoperative assessment to reduce unnecessary thyroidectomy (line 283-286).
Thank you also for your remark about molecular examination.
We also added further information about the molecular studies to the discussion (line 384-391).
In cases of indeterminate ultrasound and/or cytological results, very often patients are referred to diagnostic surgery. Supportive methods are necessary to avoid the operation and its potential complications. The first gene expression classifier was introduced to a preoperative diagnostic of thyroid nodules in the USA. Currently, the use of molecular tests is a routine clinical practice in the USA and recommended by American scientific societies. A recent study proved several genetic mutations considered markers of aggressive tumor behavior, including mutations in RAS, PIK3CA, PTEN, P53, ALK, and BRAF genes. However, FNAB is to date a reference standard in directing patients for operation and surveillance.
Round 2
Reviewer 1 Report
I appreciate the effort of author for improving the manuscript. However, it remains a significant criticism. The cancer rate they found in EU-TIRADS 5 (43%) is too different from that recorded in other previously published articles (about 85%). This finding is probably due to a selection bias. If the included in their series patients undergone thyroidectomy due to thyroiditis and Graves', this can represent the main reason. Furthermore, other causes can be present. All these cases must be excluded to reach a cancer rate of at least 70-75%. If we evaluate a nodule in a thyroiditis or Graves' in clinical practice, we carry a significant risk of false positive US assessment (these lesions cannot be perfectly evaluated, indeed). All international TIRADSs have a main objective: to reduce unnecessary FNA. Starting from this point of view, in a study of validation of a TIRADS, including atypical introduces a significant bias. I encourage the authors to exclude all potential false-positive EU-TIRADS 5 with the aim to increase the cancer rate in this class.
Author Response
Thank you very much for your kind words about the amendments, which we had already made to improve the quality of our manuscript. We also wish to thank for your additional comment about potential selection bias in classifying our lesions for EU-TIRADS category 5.
According to the Reviewer suggestion, we did additional calculation of the cancer risk in EU-TIRADS 5 category after exclusion of the patients with concomitant Hashimoto’s thyroiditis or Graves’ disease, namely autoimmune thyroid disease (AITD). The presence of AITD was defined as concentration of anti-thyroid autoantibodies (TPOAb – antithyroid peroxidase, TgAb – anti-thyroglobulin, or TRAB – anti- TSH receptor autoantibodies) at least more than three times more than upper normal range and a history of thyroid dysfunction treated prior to thyroidectomy. Additionally, we excluded patients with suspicion of primary hyperparathyroidism (PHP) on the basis of clinical and biochemical signs of hyperparathyroidism prior to surgery, to exclude the possibility of inclusion of parathyroid adenomas as lesions with category EU-TIRADS 5. The initial cancer rate in our group of patients with EU-TIRADS 5 category was 43,5%, but after exclusion of those potentially confounding factors the cancer rate was equal to 46,6%. We added the results of this calculation to the results section and discuss them in the discussion section.
The results of our re-calculation suggest that large proportion of thyroid cancers not coincide with AITD. In addition, it turned out that inclusion of patients with concomitant AITD or parathyroid adenomas, does not increase meaningfully the cancer rate in EU-TIRADS 5 category in our group. Thus, inclusion of these patients was not the reason and excluding these patients does not improve the specificity of EU-TIRADS 5 category for cancer prediction. In our opinion the classification should be universal and not only for particular group of patients; in some subpopulations, the incidence of AITD reaches even 10% i.e. in women after 60-years of age. In some patients, the thyroid autoimmune status is unknown. It is controversial to advice to measure thyroid autoantibodies to make sure that we can apply EU-TIRADS scale in patients qualified for thyroidectomy. On the other hand, authors of ETA guidelines (European Thyroid Association Guidelines for Ultrasound Malignancy Risk Stratification of Thyroid Nodules in Adults: The EU-TIRADS. Russ G, Bonnema SJ, Erdogan MF, Durante C, Ngu R, Leenhardt L. Eur Thyroid J. 2017 Sep;6(5):225-237) indicate that estimated cancer rate in EU-TIRADS category 5 should be between 26-87%. Our initial results (before we excluded patients with AITD or parathyroid adenomas) with cancer rate in EU-TIRADS 5 at the level of 43% falls exactly within this range. In fact, the EU-TIRADS scale should be used to qualify the patients for the biopsy, not constitute a single argument to send the patient for thyroidectomy. Also, presence category Bethesda 5 or 6 in small nodules, is not unambiguous with direction to the operation. The surgeon together with the patient make a decision about the treatment.
